# Human-water interface in hydrological modeling: Current status and future directions

Yoshihide Wada[1,2], Marc F. P. Bierkens[2,3], Ad de Roo[2,4], Paul A. Dirmeyer[5], James S. Famiglietti[6], Naota Hanasaki[7], Megan Konar[8], Junguo Liu[1,9], Hannes Müller Schmied[10,11], Taikan Oki[12,13], Yadu Pokhrel[14], Murugesu Sivapalan[8,15], Tara J. Troy[16], Albert I. J. M. van Dijk[17], Tim van Emmerik[18], Marjolein H. J. Van Huijgevoort[19], Henny A. J. Van Lanen[20], Charles J. Vörösmarty[21,22], Niko Wanders[2,23], and Howard Wheater[24]

[1]International Institute for Applied Systems Analysis, Schlossplatz 1, A-2361 Laxenburg, Austria
[2]Department of Physical Geography, Utrecht University, Heidelberglaan 2, 3584 CS Utrecht, The Netherlands
[3]Unit Soil and Groundwater Systems, Deltares, Princetonlaan 6, 3584 CB Utrecht, The Netherlands
[4]Joint Research Centre, European Commission, Via Enrico Fermi 2749, I - 21027 Ispra, Italy
[5]Center for Ocean–Land–Atmosphere Studies, George Mason University, 4400 University Dr, Fairfax, VA 22030 USA
[6]NASA Jet Propulsion Laboratory, California Institute of Technology, 4800 Oak Grove Dr, Pasadena, CA 91109, USA
[7]National Institute for Environmental Studies, 16-2 Onogawa, Tsukuba, Ibaraki 305-8506, Japan
[8]Department of Civil and Environmental Engineering, University of Illinois at Urbana-Champaign, 205 N Mathews Ave, Urbana, IL 61801, USA
[9]School of Environmental Science and Engineering, South University of Science and Technology of China, No.1008, Xueyuan Blvd, Nanshan, Shenzhen, 518055, China
[10]Institute of Physical Geography, Goethe-University, Altenhoeferallee 1, D-60438 Frankfurt am Main, Germany
[11]Senckenberg Biodiversity and Climate Research Centre (BiK-F), Senckenberganlage 25, D-60325 Frankfurt am Main, Germany
[12]Institute of Industrial Science, The University of Tokyo, 4−6−1 Komaba, Meguro, Tokyo 153-8505, Japan
[13]United Nations University, 5 Chome-53-70 Jingumae, Shibuya, Tokyo 150-8925, Japan
[14]Department of Civil and Environmental Engineering, Michigan State University, East Lansing, MI 48824, USA
[15]Department of Geography and Geographic Information Science, University of Illinois at Urbana-Champaign, Springfield Avenue, Champaign, IL 61801, USA
[16]Department of Civil and Environmental Engineering, Lehigh University, 1 West Packer Avenue, Bethlehem, PA 18015-3001, USA
[17]Fenner School of Environment & Society, The Australian National University, Linnaeus Way, Canberra, ACT 2601, Australia
[18]Water Resources Section, Faculty of Civil Engineering and Geosciences, Delft University of Technology, Stevinweg 1, 2628 CN Delft, The Netherlands
[19]Program in Atmospheric and Oceanic Sciences, Princeton University, 300 Forrestal Rd, Princeton, NJ 08544, USA
[20]Hydrology and Quantitative Water Management Group, Wageningen University, Droevendaalsesteeg 4, 6708 BP Wageningen, The Netherlands
[21]Environmental Sciences Initiative, CUNY Advanced Science Research Center, 85 St Nicholas Terrace, New York, NY 10031, USA
[22]Civil Engineering Department, The City College of New York, 160 Convent Avenue New York, NY 10031, USA
[23]Department of Civil and Environmental Engineering, Princeton University, 59 Olden St, Princeton, NJ 08540, USA
[24]Global Institute for Water Security, University of Saskatchewan, 11 Innovation Blvd, Saskatoon, SK S7N 3H5, Canada

*Correspondence to*: Yoshihide Wada (wada@iiasa.ac.at)

**Abstract.** Over recent decades, the global population has been rapidly increasing and human activities have altered terrestrial water fluxes to an unprecedented extent. The phenomenal growth of the human footprint has significantly modified hydrological processes in various ways (e.g., irrigation, artificial dams, and water diversion) and at various scales (from a watershed to the globe). During the early 1990s, awareness of the potential for increased water scarcity led to the first detailed global water resource assessments. Shortly thereafter, in order to analyse the human perturbation on terrestrial water resources, the first generation of large-scale hydrological models (LHMs) was produced. However, at this early stage few models considered the interaction between terrestrial water fluxes and human activities, including water use and reservoir regulation, and even fewer models distinguished water use from surface water and groundwater resources. Since the early 2000s, a growing number of LHMs have incorporated human impacts on the hydrological cycle, yet the representation of human activities in hydrological models remain challenging. In this paper we provide a synthesis of progress in the development and application of human impact modeling in LHMs. We highlight a number of key challenges and discuss possible improvements in order to better represent the human-water interface in hydrological models.

## 1 Introduction

The Earth's surface has undergone drastic changes due to the human-driven alteration of land use and vegetation patterns and the management of surface water and groundwater systems (Bondeau et al., 2007; Gerten et al., 2007; Rost et al., 2008). Over the last century, the global population has quadrupled and currently exceeds 7 billion, half of which live in urban areas. The rapidly growing population and rising food demands have caused a drastic six-fold expansion of global irrigated areas during the 20th century (Siebert et al., 2015). Human needs for water are ever-increasing, dominated currently by agricultural irrigation for food production worldwide (>70%). However, rapid urbanization and economic development are likely to be the main drivers for increasing water demands worldwide (Wada et al., 2016c). Humans extract vast amounts of water from surface water and groundwater resources (Siebert et al., 2010; Siebert and Döll, 2010; Wisser et al., 2010; Konikow, 2011), and these amounts have increased from ~500 to ~4000 $km^3$ $yr^{-1}$ over the last 100 years (Oki and Kanae, 2006; Hoekstra and Chapagain, 2007; Hanasaki et al., 2008a,b; Wada et al., 2014). Tens of thousands of artificial dams have been built in major river systems with total storage capacities exceeding 8000 $km^3$ worldwide (Nilsson et al., 2005; Lehner et al., 2011). These are used to boost water supply, to provide flood control, and to serve as a source of hydropower generation to supply the energy needs for industries (Liu et al., 2015, 2016). However, regional and seasonal variations of water supply and demand are large, causing water scarcity in various regions of the world (Gleick, 2000, 2003; Vörösmarty et al., 2000; Oki and Kanae, 2006; Kummu et al., 2010). In such regions, groundwater is often intensively used to supplement the excess demand, often leading to groundwater depletion (Rodell et al., 2009; Famiglietti et al., 2011; Konikow, 2011; Gleeson et al., 2012; Scanlon et al., 2012; Taylor et al., 2013). Climate change adds further pressure on the Earth's water resources and is likely to amplify human water demands due to increasing temperatures over agricultural lands (Dirmeyer et al., 2006, 2009, 2014; Wada et al., 2013a,b; Haddeland et al., 2014; Schewe et al., 2014).

Terrestrial water fluxes have been affected by humans tp an unprecedented extent and the fingerprints that humans have left on the Earth's water resources are increasingly discernible in a diverse range of records that can be seen in both surface freshwater and groundwater resources. The United Nations alerts us that in water scarce regions the shortage of water is beginning to limit economic growth and create large uncertainties for the sustainability of future water supply (World Water Assessment Programme, 2003). Given rising levels of human footprint, and the heavy dependence of the world economy and livelihoods on water, human impacts on land and water systems are pervasive (World Water Assessment Programme, 2016). Agriculture and urbanization affect the delivery and quality of water to river and groundwater systems (Siebert et al., 2010); many river flows are regulated (Lehner et al., 2011) and threatening ecological flows (Poff et al., 2010); water use, in particular for irrigation, can be a dominant factor in the hydrological cycle, including effects on land-atmosphere feedbacks and precipitation (Wada et al., 2016a) that can have substantial non-local impacts (Dirmeyer et al. 2009; Tuinenburg 2012; Wei et al. 2013; Lo and Famiglietti, 2013). In an era now designated as the Anthropocene (Steffan et al., 2011; Montanari et al., 2013; Savenije et al., 2014), global hydrology must therefore be treated as a coupled human-natural system.

During the early 1990s, awareness of the potential for global water scarcity led to the first detailed global water resource assessments comparing water availability with water use based on national statistics and observed climate information (Falkenmark, 1989; Falkenmark et al., 1997). Shortly thereafter, in order to analyse the human perturbation on water resources, the first generation of large-scale hydrological models (LHMs) appeared (Bierkens, 2015). These models solve the local water balance consistently across large scales and calculate river discharge by accumulating gridded runoff over a river network constructed from topographic information (Vörösmarty et al., 1989). However, at this early stage few models considered the interaction between terrestrial water fluxes and human activities, including water use and reservoir regulation, and even fewer models distinguished water use from surface water and groundwater resources (Nazemi and Wheater, 2015a,b). The phenomenal growth of the human footprint has significantly modified hydrological processes in various ways (e.g., land use, artificial dams, and water diversion) and at various scales (from a watershed to the globe) (Sivapalan et al., 2012; Sivapalan, 2015). The increasing number of recent global and regional studies show that human activities can no longer be neglected in hydrological models, since otherwise the resulting assessments will be biased towards the natural conditions in many parts of the world. Since the early 2000s, a growing number of LHMs are incorporating human impacts on the hydrological cycle; however, human representations in hydrological models are still rather simplistic.

In this paper, we review the evolution of modeling human impacts on global water resources. The paper provides a synthesis of progress in the development and application of LHMs that include an explicit treatment of human-water interactions, the lessons learned, challenges faced, and perspectives on future extensions. In this review, a number of key challenges are identified and possible improvements are discussed. This synthesis paper is an outcome of the Symposium in Honor of Eric Wood: Observations and Modeling across Scales, held June 2-3, 2016 in Princeton, New Jersey, USA. The primary objective

of this contribution is to discuss the integration of human activities into process based hydrological modeling and to provide future directions.

## 2 Evolution of representing human impacts in hydrological models

To analyse the impacts of human-induced changes on water resources consistently across large scales, a number of LHMs have been developed since the late 1990s. In the early stages, the surface water balance (e.g., runoff and evaporation) was primarily simulated in LHMs and runoff was routed down the simulated river systems (Vörösmarty et al., 1989). These calculations were then compared to population and water use data to derive the degree of human water exploitation or water scarcity primarily at an annual temporal scale (e.g., Alcamo et al., 1997, 2003a,b; Arnell et al. 1999; Vörösmarty et al. 2000; Oki et al. 2001). LHMs typically simulate the dynamics of soil moisture due to precipitation and evapotranspiration, the generation of runoff and the discharge through the river network on a coarse grid (~50-100km). Most LHMs are based on the water balance concept and track the flows of water through a number of storage including canopy, soil and groundwater. Most LHMs are not fully calibrated, but in some cases they are tuned with regional parameters (Widén-Nilsson et al., 2007).

Conceptual models are often chosen as they are deemed to be robust and parsimonious in their data requirements. In fact, for water budget calculations supporting water resource assessments, these more parsimonious models can be shown to yield similar annual and sub-annual estimates as more complex models, especially in the context of the lack of comprehensive and high quality forcing data sets (Federer et al., 1996, 2003). In recent developments, however, LHMs are becoming more physically based and process oriented with large-scale data more readily available and there is increasing incorporation of better hydrological representations for various processes including runoff generation, soil physics and groundwater representation. For example, water flows and water storages are calculated for individual hydrological components such as rivers, lakes, reservoirs, and groundwater, among others (e.g., Döll et al. 2003; Hanasaki et al. 2008a,b; Rost et al. 2008; Wada et al. 2011a,b; Pokhrel et al. 2012). More sophisticated hydrological schemes to consider seasonal difference such as in runoff, snowmelt, soil moisture, and lake and dam regulation have been implemented. Water use is now often subdivided among these different water sources into specific sectors such as irrigation, livestock, manufacturing, thermal power cooling, municipalities, and the aquatic environment (Hanasaki 2008a,b; Wada et al. 2011a,b; Flörke et al. 2013; Pastor et al. 2014). Irrigation schemes to calculate the water demand have also been improved from simply using the difference between potential and actual evapotranspiration to using a soil moisture deficit that is dynamically coupled with hydrology. Nowadays, many LHMs consider the dynamic feedback between hydrology and human water management via irrigation-soil moisture dynamics, reservoir-streamflow interaction, and water allocation-return flow (withdrawals minus consumption) dynamics (Döll et al. 2012; Wada et al., 2014; Pokhrel et al., 2015). Regional hydrological models (RHMs) consider even more complex feedback and co-evolution of coupled human–water systems (Liu et al., 2014). Many human activities, such as human induced changes in the surface and subsurface of a watershed, are not for the purpose to change the water cycle but

they indeed alter the water cycle and water resources. These impacts are increasingly accounted for in the current generation of LHMs and RHMs.

LHMs have been developed primarily to assess water resource availability and use under human land-water management practices (Arnell, 1999; Alcamo et al., 2003a,b; Döll et al., 2009, 2012; Gosling and Arnell, 2016; van Beek et al., 2011; Wada et al., 2011a,b, 2014; Wisser et al., 2010), but they are typically water balance models that do not solve the land surface energy balance (Nazemi and Wheater, 2015b; Overgaard et al., 2006), even though there were some attempts to couple land surface models (LSMs), that consider surface energy balance, with global river routing model (Oki and Sud, 1998) to estimate the availability of water resources globally (Oki et al., 2001; Hanasaki et al., 2008a,b). The primary focus in their development remains the accurate simulation of river discharge at relevant scales. To achieve this, most LHMs typically employ a few parameters that can be tuned to match the simulated discharge with observations (e.g., Döll et al., 2003; Wisser et al., 2010). The underlying assumption is commonly that since the models are tuned to capture the observed discharge, other fluxes, such as ET are automatically simulated with reasonable accuracy. However, it is well known that focussing on a single criterion such as discharge does not guarantee good performance for other fluxes (Hogue et al., 2006). LHMs are designed to be used in an offline mode with given climate information provided as an external input, and are not generally coupled with climate models (GCMs).

However, some early LHMs were developed to be incorporated as LSMs in global climate models (GCMs) or Earth System Models (ESMs) (Yates, 1997), or as stand-alone hydrological models such as VIC (Wood et al. 1992; Nijssen et al., 2001a,b) (see Table 1 for classifications). In contrast to LHMs, LSMs have been developed as the integral components of GCMs. The development of LSMs can be traced back to early work by Thornthwaite and Mather (1957) and Manabe (1969), who developed a simple "bucket model" based on the concepts of Budyko (1965). Early LSMs used simple parameterizations for solving surface energy and water balances without explicitly simulating the influence of land use change and human water management on surface hydrological processes (Deardorff, 1978; Bonan, 1995). They are used to estimate the exchange of energy, heat, and momentum between the land surface and atmosphere in GCMs, and to close budgets. Since terrestrial hydrological processes exert profound influence on the overlying atmosphere (Shukla and Mintz, 1982; Koster et al., 2004), LSMs have advanced through intensive improvements in the representation of vegetation, soil moisture, and groundwater processes (e.g., Lawrence et al., 2011) by both the atmospheric and hydrologic research communities (Sellers et al., 1997).

As a growing body of literature highlights the need to represent human activities in GCMs, studies have begun to incorporate human factors in a number of LSMs. For example, Pokhrel et al. (2012, 2015) incorporated a number of human land-water management schemes including reservoir operation (Hanasaki et al., 2006), irrigation, and groundwater pumping into the MATSIRO LSM (Takata et al., 2003), and examined the human alteration of land surface water and energy balances. A number of other studies have incorporated similar schemes in a variety of global land surface models including the

Community Land Model (CLM; Leng et al., 2014, 2015), the Organizing Carbon and Hydrology in Dynamic Ecosystems (ORCHIDEE) model (de Rosnay et al., 2003), and the Noah LSM (Ozdogan et al., 2010). Apart from these global studies, various regional-scale studies have also developed human impacts schemes to be incorporated in GCMs (e.g., Voisin et al., 2013; Ferguson and Maxwell, 2012; Condon and Maxwell, 2014).

In addition to simulating land surface hydrology, LSMs provide the lower boundary conditions for atmospheric simulations in GCMs. They typically employ sub-hourly time steps and solve the energy balance on land, which is vital to the simulation of the diurnal patterns of surface and soil temperature variations required by their parent climate models to facilitate a dynamic linkage between land and atmosphere through continuous exchange of moisture, energy, and momentum.

Considering energy balances in LSMs is crucial not only to provide the boundary fluxes to the atmospheric models, but also to simulate alteration of land surface energy partitioning due to human activities such as irrigation (Ozdogan et al., 2010; Pokhrel et al., 2012), and consequently to understand its climate impact (e.g., Boucher et al., 2004; Lo and Famiglietti, 2013; Sacks et al., 2009; Sorooshian et al., 2014). Furthermore, consideration of energy balance also makes these models suitable for coupling with agronomy-based crop models to dynamically simulate the changes in crop growth and productivity,

including stage-dependent heat stress change under climate change (e.g., Osborne et al., 2015).

Some large-scale dynamic vegetation models (DVMs) include land surface hydrology and human water management, such as the LPJmL model and JULES, as an integrated component of land use and vegetation dynamics including $CO_2$ fertilization effects (Gerten.et al., 2007; Clark et al., 2011; Konzmann et al., 2013). Notwithstanding such growing

sophistication, most current-generation of LHMs, LSMs, and DVMs still fall short of simulating the direct human influence on the terrestrial freshwater systems (Nazemi and Wheater, 2015a,b; Pokhrel et al., 2016), leaving the task of representing human land-water management activities within these models, and consequently in GCMs and ESMs, as one of the grand challenges for the hydrologic research community (Wood et al., 2011).

## 3 Current challenges of modeling coupled human-water interactions

### 3.1 Modeling human impacts on extremes

Hydrological extremes (i.e., drought and flood events) and water scarcity have become more severe over the last decades in multiple regions across the world (Hisdal et al., 2001; Lins et al., 1999; Stahl et al., 2010; Jongman et al., 2012; Di Baldassarre et al., 2017), which has led to substantial societal and economic impacts (Stahl et al., 2016; Wilhite et al., 2007). Many large-scale studies focus on drought and flood induced by climate extremes (e.g., Milly et al., 2005; Hirabayashi et al.,

2013; Orlowski and Seneviratne, 2013; Dankers et al., 2014; Jongman et al., 2014; Prudhomme et al. 2014; Sheffield and Wood, 2008; van Huijgevoort et al., 2014; Wanders and van Lanen, 2015; Wanders and Wada, 2015b); however, human water management is found to be an important factor affecting regional water supply and hydrological variability (Wada et

al., 2013a,b; van Loon et al., 2016; Di Baldassarre et al., 2017). Recent studies explicitly model human interventions (e.g., human water use and reservoir regulation), which enables attribution of the impact of droughts, floods and water scarcity to natural and human processes (Di Baldassarre et al., 2013a,b; Forzieri et al., 2014; Haddeland et al., 2014; van Dijk et al., 2013; van Loon and van Lanen, 2013; Veldkamp et al., 2015; Wada et al., 2013a,b; Wanders and Wada, 2015a; He et al., 2017).

With that said, commonly used drought indicators such as the Standardized Precipitation Index (SPI) and Standardized Precipitation and Evapotranspiration Index (SPEI) are not able to capture the human impacts that affect drought in streamflow and groundwater. For example, we argue that, instead of potential, actual evapotranspiration should be used, which allows better quantification of the impact of agricultural irrigation under increasing temperatures. Figure 1 demonstrates a significant difference in the duration of droughts in California based on SPEI with potential and actual evapotranspiration under natural conditions (Natural) and human water management (Human). Furthermore, the influence of artificial water storage such as reservoirs on hydrological extremes including drought and flood events is obvious in intensively managed agricultural regions. Without considering human water management, modeling recent severe droughts, such as the California drought, would yield a very different picture, which may be misleading for developing adaptation measures. In California, drought impacts were alleviated due to extra water available from reservoirs, at least on the short term. Irrigation return flow to groundwater storage also works in a similar manner (Figure 1). However, water use dominated by groundwater pumping led to a significant lowering of groundwater levels (Figure 1, middle right panel), emphasizing that these processes should be incorporated in state-of-the-art hydrological models. Modeling flood events without human water management would also yield a very different picture particularly in developed countries where regional water storage and dikes are prevalent for flood mitigation (Lauri et al., 2012; Mateo et al., 2014). Without considering these regional measures, flood events could be largely overestimated in hydrological model simulations.

### 3.2 Human impact indicators

Over the last few decades numerous water resources assessment indicators have been developed alongside the improvement in human impact modeling frameworks. As overuse of water resources emerged in various regions of the world, Falkenmark (1989) pioneered the concept of the Water Crowding Index (WCI) using a threshold value to describe different degrees of water scarcity. This indicator defines per country water stress based on the per capita annual renewable freshwater resources (~blue water). Annual renewable freshwater resources of 1,700 $m^3$ $yr^{-1}$ per capita are taken as the threshold below which water scarcity occurs with different levels of severity, and that 1,000 $m^3$ $yr^{-1}$ per capita as a general indication of a limitation to economic development (Falkenmark et al., 1997). While this is still one of the most commonly used indicators, this water scarcity metric has evolved into a more comprehensive, spatially-explicit and sector-specific indices including agricultural (irrigation and livestock) and industrial water needs (Alcamo et al., 1997, 2003a,b; Arnell, 1999; Vörösmarty et al., 2000; Oki et al., 2001). Many recent studies compare total water withdrawals or consumption (agriculture, industry and

households) to water availability to express the fraction of the available water taken up by demand at the finer grid level, since country-based estimates hide substantial within-country variation of water availability and demand (Hanasaki et al., 2008a,b; Wada et al., 2011a,b). Focusing on the African continent, Vörösmarty et al. (2005) emphasized the essential nature of the topology of river networks to differentiate between climatic and hydrologic water stress in macro-scale water resource assessments. In the current operational European water management and policy, the Water Exploitation Index (WEI) is used, reflecting both water consumption and withdrawals divided by water availability (De Roo et al., 2012). Water availability is local renewable freshwater with incoming streamflow from upstream parts of a river basin.

In general, a region is considered to experience water scarcity when the ratio of water withdrawal to availability is higher than 0.4 (0.2 in case of water consumption), considering the sustainability of renewable water resources. In order to track the volume of water used to produce a commodity, good or service along the various steps of production and in the international trade, Hoekstra (2009) and Hoekstra and Mekonnen (2012) pioneered the water footprint concept, which classifies and quantifies the water source, but does not assess the impact of human water use on natural stocks and flows, because it generally focuses on the volumes of water required without quantifying the volume of water available in the region. A few studies (Oki and Kanae, 2004; Oki et al., 2017) demonstrated how importing water intensive commodities such as crops and meat virtually reduces water scarcity in water crowded nations and their relationship with the economic situation of the nations. There are recent attempts to integrate both water quantity and quality in water scarcity assessment (e.g., Liu et al, 2016; Zeng et al., 2013), and water quality including water temperature is closely linked to human interactions with water systems. In recent years, various new water resources assessment indicators have been developed including the Blue Water Sustainability Indicator (BlWSI; Wada and Bierkens, 2015) that considers both renewable and non-renewable groundwater resources, and environmental flow requirements. Soil moisture (~green water) stress is still rarely assessed in the context of human water needs (Schyns et al., 2015), even though soil moisture is the major water source for global food production (~80%) (Kummu et al., 2014).

When considering water resource assessment indicators for water scarcity and drought, classical non-transient thresholds for a baseline period (e.g., 1980-2010) are often assumed for future assessments. This may not be meaningful for considering the coming decades, when humans and nature may gradually adapt to a new hydrological state arising from either climate (Wanders et al., 2015) or other more direct drivers (Vörösmarty et al., 2010). This indicates an urgent need to develop more socially and ecologically relevant indicators that connect water science to the international society. This development should be addressed within the hydrological community.

### 3.3 Modeling human impacts on groundwater resources

The first assessments of global water resources (Falkenmark, 1989; Falkenmark et al., 1997; Alcamo et al., 2003a,b, 2007; Vörösmarty et al., 2000) were mostly focused on blue water demand and availability, where the latter was assumed to be

equal to streamflow. No distinction was made between groundwater and surface water use. This distinction was unnecessary because these analyses were limited to renewable water resources and long-term averages, where streamflow also includes baseflow and it makes no difference for the budget calculations whether water is withdrawn directly from the river or from shallow groundwater pools that are in dynamic equilibrium with climate forcings. In later analyses, groundwater use was estimated implicitly (e.g. Wisser et al., 2008; Rost et al., 2008). These and subsequent assessments of groundwater use have evolved from assessments of groundwater use without hydrological feedbacks into those with feedbacks between the groundwater and surface water system: for example, via agricultural irrigation where groundwater is supplied over irrigated areas thereby affecting the surface water balance.

In the early developments, water demand is estimated first. Next, total water demand is attributed to available surface water and groundwater resources, leading to estimates of groundwater and surface water consumption, after subtracting return flows. As stated above, no specific feedbacks to the hydrological system are included. Instead, in order to obtain cell-specific blue water availability, for each model cell total upstream water consumption (groundwater plus surface water) is abstracted from the natural streamflow in post-process. Note that between these studies, very different assumptions were made about the allocation of water demand to surface water and groundwater. For example, in H08 (Hanasaki, 2008a,b, 2010), surface water is preferentially abstracted over groundwater, whereas in WBMplus (Wisser et al., 2008), water from reservoirs and groundwater is preferentially abstracted. In LPJmL (Rost et al., 2008), irrigation demand is attributed to surface water and groundwater resources using temporally invariant fractions, while in WaterGAP (Döll et al., 2012) groundwater abstractions are calculated with temporally invariant but sector- and country-specific fractions of total water demand. In PCR-GLOBWB (van Beek et al., 2011; Wada et al.,2011a,b) where local (cell-specific) groundwater abstractions are calculated by downscaling country-specific reported abstraction rates with local water demand and surface water availability.

Irrespective of the attribution approach used, these models have to deal with regions where both surface water and groundwater are insufficient to satisfy demand. The resulting water gap is either reported or is assumed to be satisfied from non-local or non-renewable water sources (Rost et al., 2008; Hanasaki et al., 2010; Vörösmarty et al., 2010), i.e., groundwater depletion or water diversions respectively. Wada et al. (2010) explicitly calculated groundwater depletion (non-renewable groundwater abstraction) using downscaled abstraction data from the International Groundwater Resources Assessment Centre (IGRAC; www.un-igrac.org) and simulated recharge. The problem with this approach, however, is that it does not correct for increased capture when calculating depletion, resulting in an overestimation of depletion rates (Konikow, 2011). De Graaf et al (2014) attempted to dynamically include groundwater abstraction into a global hydrological model. Here, attribution of groundwater abstraction is dynamic and based on the ratio of recharge to river discharge (groundwater to surface water availability). Abstractions are actually taken from groundwater reservoirs and affect surface water-groundwater interaction through baseflow and river infiltration. Return flows from irrigation, domestic and industrial water abstractions are included as well. Similar schemes were developed by Wada et al. (2014) and Döll et al (2014).

Although these schemes are able to mimic the interaction between groundwater pumping and hydrology, they lack the groundwater dynamics needed to represent the non-linear relationship between groundwater pumping and groundwater-surface water interaction. Building on a previously developed global hydrogeological schematization (De Graaf et al., 2015) De Graaf et al. (2017) recently calculated groundwater depletion with a two-layer transient global groundwater model coupled to the global hydrological model PCR-GLOBWB. In this study, they were able to account for increased capture leading to global depletion rates that are smaller than previously calculated by Wada et al. (2010) and are slightly larger than estimated by Konikow (2011).

Recently, groundwater use has also been incorporated in LSMs within climate models. A notable example is from a study by Wada et al (2016a) where the contribution of groundwater depletion to sea-level change was assessed by including groundwater withdrawal and consumption in the Community Earth System Model (CESM). Pokhrel et al. (2015) incorporated a water table dynamics scheme and a pumping scheme into the LSM called the Minimal Advanced Treatment of Surface Interaction and Runoff (MATSIRO; Takata et al., 2003) to explicitly quantify the natural and human-induced groundwater storage change. These developments provide evidence that groundwater dynamics and groundwater use are slowly but surely being incorporated in the global modeling of human impacts on the terrestrial hydrological cycle. However, it should also be recognised that available global hydrogeological schematisations (e.g., Gleeson et al., 2014; De Graaf et al., 2015, 2017) are grossly over-simplified and a joint effort is urgently needed from the hydrogeological and land surface modeling communities to improve these relatively simplistic models. Otherwise, further progress on groundwater use modeling will be seriously hampered.

## 3.4 Incorporating regional water management

It is important to note that although the influence may not be large at the global scale, urban and rural water supply infrastructure is much more diverse and regulated in many developed countries, which is not realistically accounted for in existing modeling frameworks. Seawater desalination, water diversions, and reclaimed water infrastructure are often developed to expand water supply in water scacre regions, but these human interventions in water systems are weakly integrated in LHMs. For example, given ever-increasing water scarcity, desalination is becoming a practical and established technique to produce freshwater from saline water in coastal arid regions in the world, typically countries in the Middle East (Voutchkov, 2013). All major coastal Australian cities now also have desalination options to intermittently or permanently supplement insufficient conventional supplies. It is reported that seawater desalination contributes almost 100% of the water supply for some cities including Makkah in Saudi Arabia (KICP, 2009). Due to the rapid development of seawater desalination plants in recent years, total capacity has been expanded from 3.52 $km^3$ $yr^{-1}$ in 1990 to 19.16 $km^3$ $yr^{-1}$ in 2014 (DesalData; http://www.desaldata.com).

Seawater desalination was seldom included in earlier simulation-based global water resource assessments, as it involves the production of fresh water that is unlimited by precipitation. In order to improve the accuracy of water use amounts globally, Oki et al. (2001) subtracted the equivalent volume of desalination water reported in FAO AQUASTAT from water uses (withdrawals) in their assessments. Wada et al. (2011) spatially distributed national statistics of desalination water along the grid cells nearby seashore. Recently, Hanasaki et al. (2016) proposed a novel method to include desalination in LHMs. They first identified the geographical distribution of areas utilizing seawater desalination (AUSD) from empirical rules utilizing global maps of aridity, GDP per capita, and distance from the coast. They then estimated the volume of desalination water production by combining the map of AUSD and simulated water deficit (i.e., difference between water requirement and water availability of conventional sources). They succeeded in reproducing the spatial extent of where major seawater desalination plants exist and the volumes of past production for major countries. Their future projections report that the production of desalination water in 2041-2070 would expand to 6.7-17.3 times current rates under various socio-economic scenarios. Numerous challenges remain for better representation of seawater desalination. For example, recently major desalination plants have been installed in semi-arid and humid climates, which is not well explained by the model of Hanasaki et al. (2016).

Another example is long-distance and cross-basin water diversions that provides additional water supplies. Some information is available, e.g. the Periyar Project (maximum capacity: 40 $m^3$ $s^{-1}$) and Kurnool Cudappah Canal (maximum capacity: 85 $m^3$ $s^{-1}$) in India, and the Irtysh-Karaganda Canal (maximum capacity: 75 $m^3$ $s^{-1}$ in Central Asia (World Bank; http://www.worldbank.org/; UNDP; http://www.undp.org). Recently, the world largest inter-basin transfer scheme, the South-to-North Water Diversion (SNWD) project, became operational and Beijing began to receive fresh water from the Yangtze River in China's south, which covers a distance of more than 1,000 kilometres (Barnett et al., 2015). These water diversions play a role in mitigating regional water scarcity, but also influence water balances in sourcing and destination basins (Zhao et al., 2015). However, artificial diversion networks and the actual amount of water transferred are difficult to parameterize, and are not represented in the current generation of modeling frameworks. Extensive urban water supplies and waste water networks are also important aspects given that half the world population currently lives in urban areas. Further efforts are needed not only for modeling but also for comprehensive data collection of global seawater desalination, water diversion, and urban water network development.

Although desalination and inter-basin water transfer are emerging examples and likely more important in the near future, regional water management is much more complicated. Current LHMs also lack dynamic trade-offs among irrigation water supply, flooding control and hydropower production, water competitions between upstream and downstream users (Munia et al., 2016; Veldkamp et al., 2017), and deficit irrigation and rainwater harvest (Döll et al., 2014). These processes are increasingly important for regional hydrological model simulation. For example, considering regional deficit irrigation practice can reduce the water demand by 30% (Döll et al., 2014), while current LHMs predominantly use optimal irrigation

practice in their model simulation. This is similar to the need to account for return flows from industry and households after water withdrawals. Water recycling and waste water treatments are becoming important mitigation measures for regional water scarcity. Modeling water recyeling and waste water treatments should be combined with local water quality information, which can provide more accurate information of absolute availability of usable water for different purposes such as drinking water, industry, and agriculture.

## 3.5 Representing land use change and rapid urbanization

Humans have transformed natural vegetation to anthropogenic land cover such as agricultural lands and pasture over 40% of the global land area (Klein Goldewijk et al., 2011; Sterling et al., 2013). Human induced land use change has profound impacts on global and regional hydrological cycle by changing the rate of evapotranspiration, runoff, and groundwater recharge, which in turn affects regional precipitation patterns and inflows to oceans (Gordon et al., 2005; Halder et al., 2016; Puma and Cook, 2010; Renner et al., 2014). Human transformation of global land cover (excluding irrigated agriculture) generally decreases evapotranspiration and increases runoff (Gordon et al., 2005). Many LHMs include the impacts of land use change, however, the land use representation in the model tend to be statically prescribed as an input parameter, while dynamic change in historical land use is a lesser focus. Compared to LHMs or LSMs, DVMs have better representation of land cover change, while land surface hydrology is treated rather simply.

Among different land use changes, urbanization is of specific interest in recent impact studies, e.g., with the focus of flood risks, hazards and vulnerability (Güneralp et al., 2015; Muis et al., 2015; Sampson et al., 2015; Tanoue et al., 2016; Winsemius et al., 2013). At present, more than half of the world's population lives in urban areas and rapid urbanization is taking place over many developed and developing regions of the world (Klein Goldewijk et al., 2011). Nevertheless, urban areas and their impact on the hydrological cycle (e.g., Jacobson, 2011) are not well represented in LHMs, mostly due to their small proportion of the global land area (Wood et al., 2011). Although the impact of urban areas to the water cycle may be local, the distribution of such areas is of high importance, e.g., for heat island and urban flood modeling (Yang et al., 2011). Among LHMs, WaterGAP uses a static input map with the percentage of impervious areas at a grid and assumes that 50% of precipitation over those areas directly reaches the surface water bodies (Müller Schmied et al., 2014). The LISFLOOD water resources model (De Roo et al., 2000) uses sub-grid fractions of urban, forest, open water, and several other land usages within the 0.1 degree (global) or 5km by 5km grid scale (for Europe) to represent the effects of land use. Several (soil) hydrological processes are consequently simulated separately (De Roo et al., 2012). Figure 2 shows the percentage of urban area at $0.5^{o}$ grid based on MODIS urban land cover classification for the year 2003. However, scale issues arise for urban land cover due to the fact that the effect of limited urban areas on the water cycle can be diminished at a large grid cell (Warburton et al., 2012) and coherent scaling relationships are missing (Reyes et al., 2016). However, satellite mapping of urban or impervious areas is improving recently (Lopez and Maxwell, 2016; Wohlfart et al., 2016; Yang et al., 2003) using the Moderate Resolution Imaging Spectroradiometer (MODIS) satellite images (Schneider et al., 2009).

A recent study shows the challenges of including small scale urban hydrological modeling (Reyes et al., 2016). However, representing urban areas as sub-grid variability and upscaling the effect of urban areas to the larger hydrological cycle may be possible (Krebs et al., 2014). For example, model simulation with and without urban areas and associated hydrological balance can be compared in urbanized catchments to see the impacts and their validation with available observations (e.g., runoff and evapotranspiration). Here, the percentage of runoff that is generated over the impervious areas may be validated and tuned to generalize the concept.

In order to better represent urban impacts on the regional hydrological cycle, more accurate assessments of urban water withdrawals and consumption are vital (Flörke et al., 2013; Wada et al., 2016b,c). Finer spatial scale population and socio-economic data are required worldwide; however, these data are typically provided at a country scale or a 0.5° grid. This leads urban water demands and supply to be geographically mismatched in current large-scale water resources assessments, and associated water scarcity and groundwater depletion are not well represented (e.g., Döll et al., 2014; Wada et al., 2014). McDonald et al. (2014) included the source of urban water supply, which led to improved water scarcity assessments. Considering rapidly increasing urban population, the model representation of urban hydrology and water management needs to be urgently considered.

## 4 A look forward

### 4.1 Modeling human activities at multiple spatial scales

Local human behaviour is an important part of the hydrological system as humans are not just external drivers or boundary conditions in hydrological systems (Sivapalan, 2012, 2015; Montanari et al., 2013; Troy et al., 2015a; van Loon et al. 2016). The field of socio-hydrology is focused on understanding the processes that link humans and water in a coupled hydrological-social system (Sivapalan et al., 2012, 2014). Socio-hydrology has emerged relatively recently as a discipline that addresses the intersection between human and natural systems (e.g., Sivapalan et al., 2012; Gober and Wheater, 2015). The basic concepts of socio-hydrology align well with the mainstream of coupled human and natural large-scale modeling efforts that have rapidly developed since the late 1990s, as discussed earlier in this manuscript (e.g., Alcamo et al., 1997, Vörösmarty et al., 2000; Oki et al., 2001; Döll et al., 2003). However, a main difference of socio-hydrology from the large-scale human impact modeling is to link bi-directional feedbacks between hydrological processes and local human behaviour, similar to agent-based modeling (ABM). Thus, socio-hydrology can be seen as a new development in human impact modeling but, so far, primarily focused on a local to regional scale, and still requires more detailed parameterizations of human behaviour and process-oriented modeling frameworks.

Socio-hydrological studies can be divided into (1) historical studies, (2) comparative studies, and (3) process-based studies. For example, as a historical study, Pande and Ertsen (2014) investigated complex cooperative agreements from ancient societies, and found that it was in fact water scarcity that triggered cooperation. For a more recent example, Kandasamy et al. (2014) revealed a "pendulum" swing in the Murrumbidgee River Basin, where population first increased, driven by agricultural development, and later decreased, driven by environmental restoration being more favoured over agriculture. In recent years several socio-hydrological models have been developed (Blair and Buytaert, 2015; Troy et al., 2015a). Di Baldassarre et al. (2013a,b) and Viglione et al. (2014) developed a conceptual "toy model" that explores the dynamics of a floodplain as a coupled human-water system. They demonstrated the relationships between the hydrological and social cycles, as human settlements in floodplains are threatened by flooding. Based on this it was revealed how societal memory of historical floods determines the (re)settling rate, and whether a society is economically growing or recessing. Several large river basins have been studied extensively, such as the Murrumbigee River Basin (van Emmerik et al., 2014), the Kissimmee River Basin (Chen et al., 2016), and the Tarim basin (Liu et al., 2014), yielding new insights into the governing hydro-social processes and relations that operate in these coupled systems. To go beyond single case studies, Elshafei et al. (2014) developed a generic framework for socio-hydrological modeling of agricultural catchments. Although the application to two Australian catchments was insightful, it remains challenging to link human and hydrological processes across multiple spatial scales over different geographies. The launch of socio-hydrology offers a new paradigm that enables us to evaluate the co-evolution of human activities and hydrology, driven by two-way feedbacks between humans and water systems over long time horizons, which was not fully addressed in the large-scale human impact modeling efforts.

Besides new opportunities and new insights, socio-hydrology can also be seen as a wicked problem (Levy et al., 2016). Human reactions to hydrological extremes can be contrasting (Loucks, 2015), and there are no widely accepted laws yet for human behavior in coupled systems (Sivapalan and Blöschl, 2015; Garcia et al., 2016). This leads to model developers deriving relations and identifying governing processes individually for each case study. Many socio-hydrological models consist of coupled differential equations that capture the dynamics of the studied system. However, it is unclear whether this is because of over-parameterization or mathematical correctness (Troy et al., 2015a; Mount et al., 2016). Either way, it is time for socio-hydrology to move beyond individual case studies, and find generalized, but locally relevant descriptions of changes in the (large-scale) human-water system (McMillan et al., 2016). Importantly, a recent study has presented a generalized socio-hydrology model of water resources and trade (Dang et al. 2016), which also highlights the opposite challenge in socio-hydrology model development, e.g. no explicit spatial representation in many economics models.

Ways forward for socio-hydrology include testing model structures and frameworks on multiple case studies, or upscaling their model boundaries and increasing the modeled system scale, and using new data, information sources, and modeling environments. Here lies the confluence where socio-hydrology models and global (hyper-)resolution models (Wood et al., 2011) might benefit from each other. Many LHMs nowadays incorporate human water management, but as discussed earlier

large uncertainties remain in model simulations (Döll et al., 2016). However, it should be noted that many recent studies report that including human influences in regional hydrology improves model performance in simulating river discharge or groundwater storage (Wada et al., 2015; Wanders and Wada, 2015a,b). For example, Ying et al. (2017) applied an ensemble of global model outputs with regional water management practices in the Yellow River basin, which yielded better surface water availability among the sub-river basins. This type of offline coupling of global models with regional water management information will facilitate the use of global models for regional application. In addition, further improvement in modeling human impact processes is crucial for realistic hydrological predictions.

Implementing local socio-hydrology models in large-scale hydrological models should be done with care, as it is important to be mindful of the temporal and spatial scales used. Human-decision making is generally modeled on a yearly basis, or lumped together as collective social structures. Integrated assessment models (IAM) such as Global Change Assessment Model (GCAM) which combine economy, energy, agriculture, climate and water resources assessment with long-term policy development can also provide a good opportunity for studying the intersection between human and natural systems at a large scale system (Hejazi et al., 2013a,b, 2014). Socio-hydrological modeling should be either done on the smallest scale (Pande and Ertsen, 2014), or on the largest societal and environmental scale (society and climate) (Ertsen et al., 2014). This is also crucial for later calibration and validation, as these should keep pace with the increase in spatial model resolution to resolve the relevant processes (Melsen et al., 2016). There should be a coordinated way forward for socio-hydrology and global (hyper-)resolution modeling efforts. Incorporating human activities globally as an endogenous factor will provide material for comparative studies for the socio-hydrological communities, increased model realism in LHMs, and better predictions of the co-evolution of the coupled human-water system.

## 4.2 Global models for regional use

Global models are specially designed for application on the global domain. They use boundary conditions and parameters that can be derived only from globally available data sets and use a limited number of robust parameters that can be used without formal parameter calibration. However, global models have recently been used for many regional applications, which requires careful attentions to how to set up global models for specific regional case studies. A straightforward approach is to run a global model for the global domain with a standard setting, and focus on analysis of the results for some specific regions. Biemans et al. (2013) used the LPJmL model (Biemans et al., 2011; Rost et al., 2008) to study future irrigation and food production in the Indian subcontinent under climate change. In their simulations, basic settings were identical to the global simulation (e.g., the spatial resolution was 0.5° by 0.5° or 50 by 50km at the equator). Earlier work by Vörösmarty et al. (1998), highlighted problems of re-scaling global water balance models to sub-global domains, using the data-rich United States as an example, revealing the numerical "penalties" of data incongruities and model formulations that would eventually be encountered in fully global scale analysis.

An advanced approach is to increase the spatial resolution of global models to better represent the regional details. Wada et al. (2016b) applied the PCR-GLOBWB model at the spatial resolution of 0.1° by 0.1°. Some models allow users to set the spatial domain and resolution freely. Mateo et al. (2014) applied the H08 model (Hanasaki et al., 2008a,b) to the Chao Phraya River in Thailand at the spatial resolution of 5' by 5'. Unlike the above mentioned global studies, they tuned several

important hydrological parameters at major river gauging stations by collecting historical meteorological and hydrological data. They succeeded in reproducing the historical long-term river discharge of the basin, including the operation of two major reservoirs and the areal expansion of inundation for a large flood event in 2011. Hanasaki et al. (2014) extended their model to quasi-real time simulation for possible application for flood monitoring in the Chao Phraya River. Masood et al. (2015) applied the model to the Ganges, Brahmaputra, and Megna Rivers in South Asia. The Australian Water Resources

Assessment (AWRA) system (van Dijk and Renzullo, 2011) couples daily time-step catchment and groundwater balance models at 0.05° resolution with a (regulated) river and reservoir model. It is used operationally by the Bureau of Meteorology to produce regular water resource assessments and water accounts (www.bom.gov.au/water/). Gosling et al. (2016) compared the simulated results of river runoff for eight large river basins in the world by using an ensemble of global to continental LHMs and an ensemble of regional catchment-scale hydrological models. The two types of model at different

spatial scales showed similar trends for the effects of global warming, indicating the possible application of LHMs for regional use. Either way, i.e. increasing spatial resolution of global models or applying global models for a specific region or catchment with fine resolution) potentially removes the barriers between regional and global models (Hattermann et al., 2017). However, ongoing efforts towards better representation of regional details are required, which would eventually improve both global models and fine scale simulation.

**4.3 Need for model intercomparison**

Modeling human behaviour is highly uncertain, but the use of a single hydrological model is still valuable to test a hypothesis, provided it is succeeded by a multi-model analysis to examine the full range of possible human impacts and model uncertainties (Tallaksen and Stahl, 2014; van Huijgevoort et al., 2013, 2014). A number of model inter-comparison projects on large-scale models have been performed (e.g., GSWP1, GSWP2, WaterMIP, and ISIMIP), and the strengths,

weaknesses, and characteristics of individual models have been compared. The focus has been on the historical energy and water balances over land (Dirmeyer et al., 2006; Douglas et al., 2006), water balance and river discharge of the past (Oki et al., 1999; Haddeland et al., 2011), and future (Hagemann et al., 2013; Schewe et al., 2014), as well as water use (Wada et al., 2013a,b, 2016c).

One of the model components that inter-comparisons have not addressed is the operation of dams. About 50,000 dams have been constructed globally (Lehner et al., 2011) and some models explicitly simulate the operation of major dams in the world (Hanasaki et al., 2006; Biemans et al., 2011; Wada et al., 2011). Masaki et al. (2017) was the first to compare the simulation results of reservoir operations of five large-scale hydrological models. They used the retrospective multi-model

simulation dataset of the ISIMIP 2a project (https://www.isimip.org/) and focused on the reservoirs of the Missouri and the Colorado Rivers in the USA. Although all of the models adopted similar algorithms of reservoir operation and used harmonized meteorological and geographical data, there were considerable differences among them. They analysed the results of only two rivers in the USA; a more systematic inter-comparison is needed that covers other regions of the world. It should be also noted that for validation of reservoir operations, data including inflow, outflow, and actual reservoir volume are not readily available worldwide, often due to political sensitivity.

### 4.4 Observing and sharing information on human water management

As mentioned several times throughout this manuscript and elsewhere (Lawford et al. 2013; Harding et al. 2014; Fekete et al. 2015), there is a serious lack of comprehensive data required to adequately constrain and evaluate hydrological models over continental to global scales. The data gaps limit our ability to fully assess model accuracy for the past, and hence to develop reliable models to predict the future. While relatively more reliable data for some hydrologic variables, such as precipitation, air temperature, and river discharge are available for many regions, data on groundwater and human water use are particularly lacking. Regional groundwater datasets are now becoming increasingly available (e.g., Scanlon et al., 2006; Fan et al., 2013) but significant challenges still remain in collecting and synthesizing data with global coverage because even the available data for most regions are not easily accessible (e.g., Hannah et al., 2011). Vast amounts of soil and aquifer analyses, including hydrogeological frameworks and measurements have been made, but the data remain dispersed and unstructured in the scientific literature, government archives, and online repositories. It is therefore essential to make community-driven efforts to compile these scattered data sets into a comprehensive Hydrogeological Information System easily accessible to the modeling community (Fan et al., 2015). Some of the available global data sets include FAO AQUASTAT for water use database, IGRAC groundwater data, the Global Runoff Data Centre (GRDC) for river flow, and the International Commission on Large Dams (ICOLD) reservoir data, but data often require substantial re-vetting and interpretation to be used for modeling studies (Lehner et al., 2011), and commonly lack information on operating rules. The hydrologic modeling community has benefited considerably from coordinated data collection and distribution efforts in the past, but it is time to revise these datasets to meet the growing need for more comprehensive, spatially explicit, time-varying data on human interactions with the hydrological cycle (Gleick et al., 2013).

Recently, use of remote sensing has provided an unprecedented opportunity to fill the spatial and temporal gaps in ground-based observations for large-scale modeling. For example, the data obtained from the Advanced Very High Resolution Radiometer (AVHRR), the Landsat mission, and the Moderate-Resolution Imaging Spectroradiometer (MODIS) have provided a unique opportunity to derive human transformed land use information. For example, MODIS data have been utilized to derive global ET at very high spatial resolution (Mu et al., 2011 Tang et al., 2009; Zhang et al., 2010), which can be used for the evaluation of global and regional irrigation impacts. The Shuttle Radar Topography Mission (SRTM) provides high resolution topography data useful for global and regional water transport and groundwater modeling. Satellite

radar altimetry and laser altimetry have provided measurements that can be used to derive water surface elevation of lakes and man-made reservoirs (Gao, 2015). The Tropical Rainfall Monitoring Mission (TRMM) delivers high resolution rainfall data for mid- and low-latitude regions for climate forcing.

In recent decades, satellite observations, such as by the Gravity Recovery and Climate Experiment (GRACE) satellite mission (Tapley et al., 2004) have further advanced our ability to better monitor the continually evolving surface and groundwater systems especially in relation to the changing climate and growing human interventions (Famiglietti et al., 2015; Lettenmaier and Famiglietti, 2006). GRACE data have been used to infer the changes in terrestrial water storage over large regions and have been widely used to study human-induced changes in surface and groundwater storages (Rodell et al.,
2009; Strassberg et al., 2009; Scanlon et al., 2012; Longuevergne et al., 2010; Famiglietti et al., 2011; van Dijk et al., 2014). The Global Precipitation Measurement (GPM), Soil Moisture Active Passive (SMAP), and Surface Water and Ocean Topography (SWOT) mission are expected to provide better information of how human activities affection terrestrial water fluxes.

Satellite observations have enabled us to better constrain and evaluate human activities in hydrological models (Famiglietti et al., 2015). This is of particular interest for less-gauged basins where conventional data are scarce. Several studies have demonstrated the use of combinations of available remote sensing products to force, calibrate and or validate hydrological models to increase the understanding of the hydrological behaviour, and the influence of human activities (e.g., Winsemius et al., 2009). However, there are inherent uncertainties and limitations in satellite-derived products (Fekete et al., 2015).
Satellite data usually provide global coverage filling the spatial gap in ground-based observations, but their temporal coverage may be limited. In addition, satellite-derived products can contain significant uncertainties because certain algorithms have to be used to derive the desired geophysical product since satellites typically measure the surface characteristics of the Earth rather than the geophysical variables themselves. Therefore, it is important to maintain ground-based observational networks in parallel with the advancements in remote sensing technology because the satellite-derived
products need to be verified with independent observations (Famiglietti et al., 2015). In fact, the TRMM Multi-satellite Precipitation Analysis (TMPA) combines products from multiple satellite and ground observations from the Global Precipitation Climatology Centre (GPCC) (Huffman et al. 2007). Recent studies also evaluated the consistency between the pure satellite-based measurements (TRMM) and TMPA at regional scale (e.g., Villarini 2010) and global scales (e.g., Zhou et al. 2014).

**4.5 Linking human impact modeling into policy development**

Given that human impacts on land and water systems are pervasive, a basic requirement for hydrological science to support local, regional and global policy is to deliver 'real-world' ESMs that incorporate the more important physical controls

associated with human influences, e.g. land use, dams, irrigation (Wheater and Gober, 2015). These are needed to support decision making at multiple scales, from local scale impacts of agricultural land management and urbanization to global-scale analysis and prediction of Earth system change, including land-atmosphere feedbacks and land-ocean freshwater delivery. Human impacts are most readily understood and represented in local-scale models, where for example process based models have access to local information on physical infrastructure, water demands and allocation rules. However, important challenges remain at that scale, for example representation of impacts of agriculture on runoff and water quality (e.g., nutrition, salinity, and pesticide). At larger spatial domains, including large river basins and transboundary waters, representing even these basic effects of human activities becomes challenging (De Lange et al., 2014). For example, data on physical infrastructure are limited at these scales, operational rules are often unknown, and while information on water allocations may or may not be available, actual water use generally has to be estimated. Nazemi and Wheater (2015 a,b) discuss the needs for new data, satellite observational tools, models and comparative analyses, as well as enhanced global coordination, to address these issues. It is evident, however, that the representation of human impacts includes not only data on physical infrastructure but also societal and cultural behaviour.

To take a simple example, operational policies for water infrastructure may not be known to downstream users, yet may have a large impact on downstream flows, and water use (as opposed to allocations), will depend on governance structures and user decisions. It therefore follows that there is a set of more complex needs for management and policy, which includes societal behaviour. It is perhaps obvious that societal behavior is an integral aspect of both policy and operational water management, but it is also important to recognize that, just as geomorphological processes influence the long term evolution of the water environment, so do human actions. As described earlier in the case of the Murrumbidgee River Basin, co-evolution of human-water system led to a government action that bought back water rights for the environment, invested in improved water use efficiency and increased environmental protection, so that environmental health is returning and water use is retreating downstream. The authors ask – could this have been predicted, and state that 'prediction of water cycle dynamics over long timescales is not feasible without including the interactions and feedbacks with human systems' (Wheater and Gober, 2015). So, for example, as society attempts to manage uncertain risks from environmental change, recognizing the non-stationarity of climate (Milly et al., 2008), it is equally important to address the non-stationarities associated with land and water management.

As we expand to larger spatial scales, many water scarce regions start to rely on external water transfers, including water diverted from other basins and virtual water from other regions via international trade, to alleviate local water problems (Hejazi et al., 2014; Zhao et al., 2015). Globalization, water diversion and virtual water also have far-reaching effects on regional water use, and hydrological cycles (Pande and Sivapalan, 2016). Hydrological models do not thus far have a capacity to capture the role of these tele-coupling water management systems. Coupled hydro-economic models are therefore needed to understand the effects of human behaviour in one place on the water systems in another place.

As a final point in this discussion of the importance of human impact modeling for policy, we suggest that a further dimension of coupled human and water systems (Gober and Wheater, 2015) concerns communication and stakeholder engagements. In commenting on the flood-plain example, Gober and Wheater (2015) note that 'The concept of social memory does not, however, adequately capture the social processes whereby public perceptions are translated into policy action, including the pivotal role played by the media in intensifying or attenuating perceived flood risk, the success of policy entrepreneurs in keeping flood hazard on the public agenda during short windows of opportunity for policy action, and different societal approaches to managing flood risk that derive from cultural values and economic interests.' This limited example illustrates that there is a rich agenda to better understand human-water interactions as a guide to policy development and implementation. More generally, Gober and Wheater (2015) note the general failure to link science with policy and associated needs for two-way iterative engagement between producers and users of scientific information to build trust and better understand the needs of policy makers and other users, and what scientists can provide to assist policy making. This could include public engagement; for example, public attitudes can be an important factor in political decisions relating to societal values associated with water management, such as the trade-offs between human water use and environmental flows.

**5 Conclusions**

This paper builds upon contributions from previous modeling efforts aimed at incorporating human activities in hydrology and in large-scale water resources assessments, and has tried to highlight the need for further improvements, including a number of key unsolved questions. To further advance the current generation of hydrological models, we have explored the possibility of including different modeling aspects of coupling human-water systems to hydrological models. The outstanding issues and shortcomings of previous large-scale water resources assessments can be grouped into five major themes: (1) issues related to current human impact modeling and associated indicators, (2) issues related to the limitations representing regional water management, (3) issues related to the need for modeling the co-evolution of human-water system, including land use and climate interaction, (4) issues related to the need for a nested approach integrating human behavior (bottom-up) into large-scale modeling (top-down), and (5) issues related to the lack of human water management information,. These five themes make up the current major challenges for the human-water interface in hydrological modeling that need substantial progress in the coming years. Despite the various limitations identified, current modeling frameworks have advanced significantly beyond earlier modeling work by accounting more realistically for human activities and the associated impacts on the terrestrial water system. Further progress in the modeling of coupled human-water system at a range of spatial scales will be important milestones not only for the hydrological science community but also for the climate and Earth system science communities. The future of human impact modeling as outlined in this paper offers a valuable opportunity for the hydrologic research community to become a truly interdisciplinary and influential Earth science than ever before.

**Acknowledgements**

This study is an outcome of the Symposium in Honor of Eric Wood: Observations and Modeling across Scales held at Princeton University during June 2-3 in 2016. The authors thank productive discussion during the symposium, which

substantially contributed to this work. We thank Eric Wood for his lifelong devotion and guidance towards global efforts to improve the understanding and practical relevance of hydrological science. We wish to thank Qiuhong Tang, Tian Zhou, and Pat Yeh for their thoughtful comments and constructive suggestions, which substantially improved the quality of the manuscript.

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

**Table 1: Type of models used to simulate global hydrology**

**Large-scale Hydrological Models (LHMs)**

- A detailed representation of terrestrial hydrological processes at long temporal (e.g., decades) but fine spatial resolutions (e.g., 10-50km)

- Inclusion of human-induced change (e.g., human water use and reservoir regulation)

e.g., H08 (Hanasaki et al., 2008a,b), PCR-GLOBWB (Van Beek et al., 2011; Wada et al., 2014, 2016), WADMOD-M (Widén-Nilsson et al., 2007), WaterGAP (Alcamo et al., 2003a,b; Döll et al., 2003), WBMplus (Vörösmarty et al., 2000; Wisser et al., 2010)

**Land Surface Models (LSMs)**

- A simplified treatment of the surface hydrology associated with human-induced change

- A focus on the interactions of the land-atmosphere for climatic simulations in global climate models (GCMs)

e.g., VIC (Wood et a., 1992), NOAH (Ek et al., 2003), MATSIRO (Pokhrel et al., 2012), JULES (Clark et al., 2011), DBH (Tang et al., 2007)

**Dynamic Vegetation Models (DVMs)**

- A simplified treatment of the surface hydrology and human land use change

- A special treatment on biosphere that enables quantitative assessment of transient changes in

vegetation and land surface hydrology in response to variations in climate and anthropogenic $CO_2$ increase

e.g., LPJmL (Gerten et al., 2007; Konzmann et al., 2013), JULES (Clark et al., 2011)

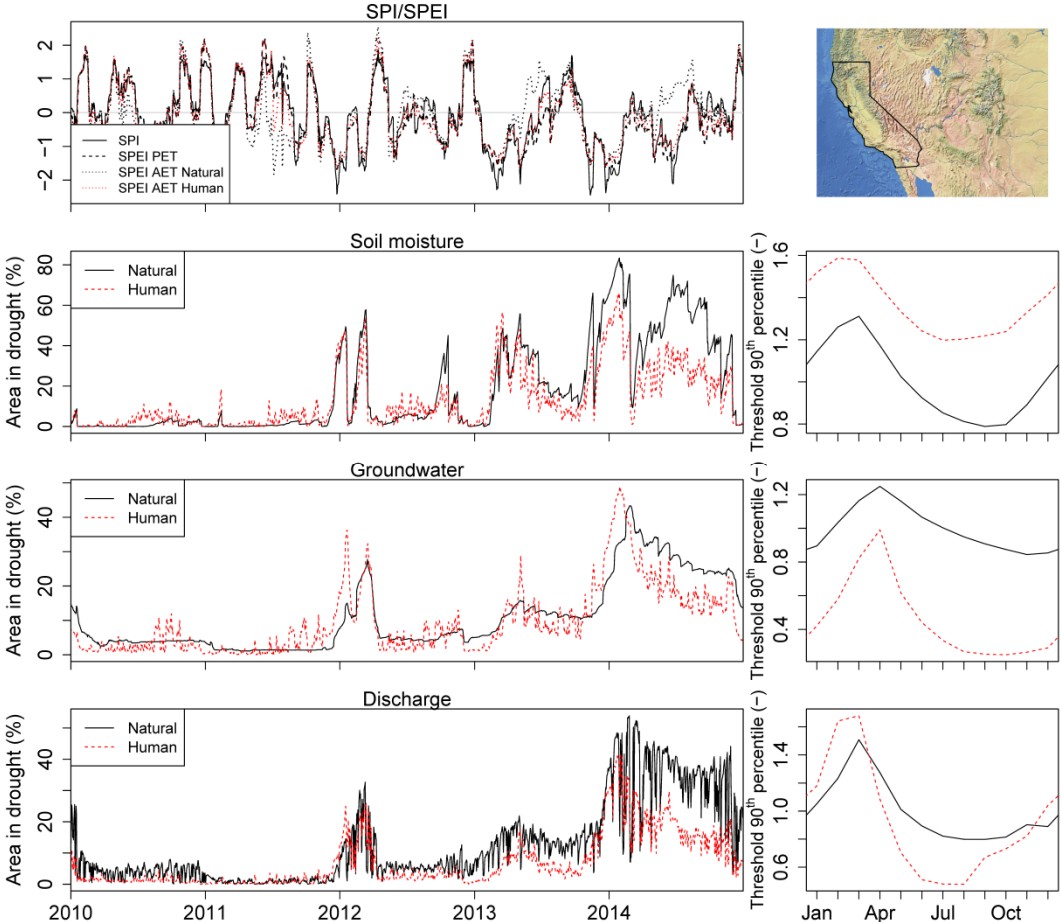

**Figure 1: Area In Drought (AID) in California (CA), USA, for the period 2010-2015. The global hydrological model PCR-GLOBWB (Wada et al., 2013a,b, 2014) has been used to simulate actual evapotranspiration, soil moisture, groundwater and river discharge at a grid of 10km by 10 km resolution. Groundwater is represented with a linear reservoir model only. We refer to Wada et al. (2014) for the detailed descriptions of model parameters and simulation. The monthly Standardized Precipitation Index (SPI), monthly Standardized Precipitation Evaporation Index with Potential Evapotranspiration (SPEI-PET), SPEI with Actual Evapotranspiration under natural and human influenced conditions (SPEI-AET natural, SPEI-AET human) were determined at the state-level. The model simulations were used to derive locally the 90th percentile variable threshold, which has been used to calculate the AID aggregated to the state-level for each hydrological variable of soil moisture, groundwater and river discharge. The 90th percentile threshold has been commonly used in drought identification (Wada et al., 2013a,b; Wanders et al., 2015) and this threshold was calculated separately for the natural situation and for the human-affected simulation shown in the right panels. All thresholds are standardised by the annual mean threshold of the natural situation.**

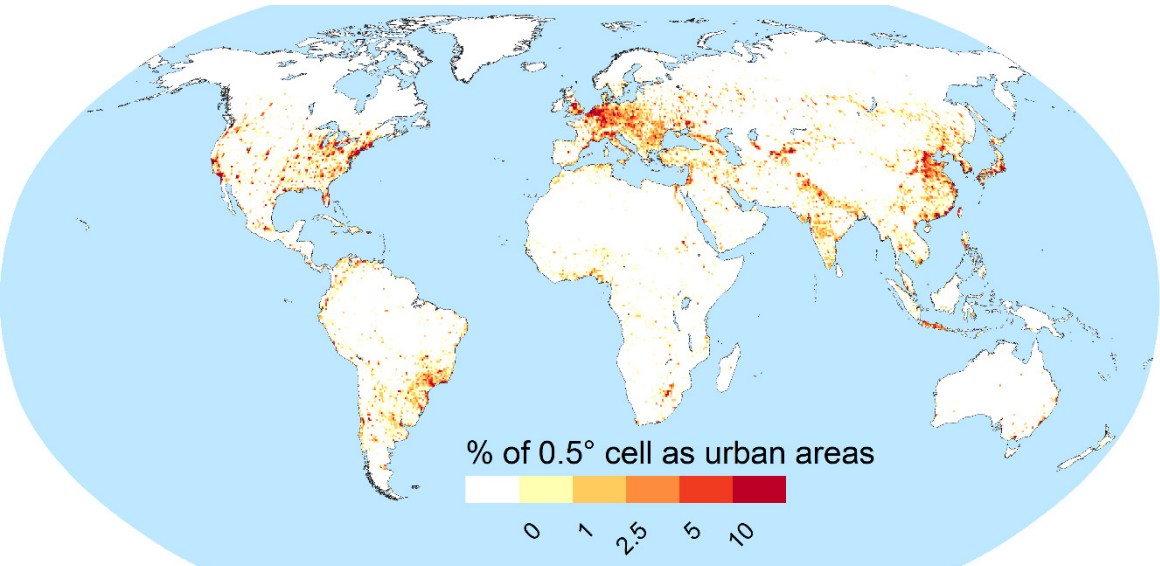

**Figure 2: MODIS urban land cover as percentages of 0.5o grid cell for the year 2003 (IGBP-classification system, class 13). The calculation was done with resampled land cover type of 0.025o tiles (2.7 x 2.7 km at the equator) due to technical reasons. Hence,**
10 **urban land cover has to be dominant in a sub-grid in order to be taken into account for 0.5o grid urban percentage. The assessment of the whole time series of MODIS land cover data (yearly data 2003-2013) shows a very robust classification, implying that during that decade and using the resampled information, not much change is detected (maximum difference is 1.2% among the years).**

