# Peer review of "Human-water interface in hydrological modeling: Current status and future directions"

_Hydrology and Earth System Sciences, 2017_

## Referee Comment (RC1) · Q. Tang (Referee) · 21 May 2017

Review Comments "Human-water interface in hydrological modeling: Current status and future directions" by Wada et al.

This is a comprehensive review on human impact modelling in large scale hydrological modelling. It is generally well written and it may be accepted for publication after minor revision.

General comments: The manuscript focuses on large scale or even global scale hydrologic modeling. If looking at other scales, the whole figure might be different. This can be reflected by adding 'large scale' or 'global scale' in the title.

Although the manuscript intends to cover human-water interface, only one aspect, i.e.

[Figure]

human water use, was reflected. The main body of text is about human impact modelling rather than human-water interactions. Human-water interface is more like an important part of future directions.

The Future Directions section (section 3) is actually the main body of the review with some points of future direction at the end of each sub-section. The Forward and Conclusion sections (section 4 and 5) provide other kind of future direction and major challenges. It might re-organized to put review (current status) in one section and future direction in another separated section.

Specific comments: The consideration of human impacts in LHM, LSM and other types of models was nicely reviewed in section 2 (Evolution of human impact modeling). However, how the representation of human impacts has been improved during the past several decades is not outlined. At the beginning, there may be only water budget to consider mean annual amount of water withdrawals. Eventually, there will be more sophisticated scheme to consider seasonal difference, to consider not only amount of water resources, but also hydropower generation, flood risk reduction, and food production. Many human activates, such as human induced changes in the underlying surface of a watershed, are not for the purpose to change the water cycle but they indeed alter the water cycle and water resources. These impacts are not well demonstrated and may be included in the discussion.

The drought and water scarcity are well reflected in sub-section 3.1 (Modeling human impacts on extremes). However, flooding is even largely affected by human impacts due to large dams and flood control project, and deserves more coverage in this important paper.

Although desalination, interbasin water transfer and urban water network are very important aspects of regional water management and were well reviewed in sub-section 3.4 (Incorporating regional water management), regional water management is much more complicated. There are lots of regional water management practices, e.g., tradeoffs between irrigation water supply, flooding control and hydropower production, water competitions between upper stream or downstream, rainwater harvest, which were not considered in many LHM yet. Some attempts and future possibilities can be discussed.

Page 2 Line 21, 'km3 yr-1'should be 'km3 yr-1'. There are also some other similar formatting errors in the manuscript.

Page 3 Lines 6-13, It is repeating the message given in the previous paragraph.

Page 3 Line 14, It is hard to understand the statement: 'water must therefore be considered as a coupled human-natural system'. Water should be an element of the coupled system, or the global water system is a coupled system?

Page 3 Line 16, Reference of 'the first detailed global water resources assessment' is required here.

Page 4 Line 25, Few model has actually considered human-water feedback. How water can affect human society and how it was represented in the model? There are some regional works under the general idea of social-hydrology, such as Liu et al. (2014, HESS, Socio-hydrologic perspectives of the co-evolution of humans and water in the Tarim River basin, Western China: the Taiji–Tire model) among others.

Page 7 Line 7, 'Human impact modeling and indicators' As other sub-sections gives one aspect of 'Human impact modeling', the title of this sub-section may focus on indicators only rather than 'modeling and indicators'.

Page 8 Line 12, Is 'The first assessments' the same as 'the first detailed assessment' in Page 3 Line 16? The references here are not in the early 1990s.

Page 16 Line 16, Besides applying global model with high spatial resolution data, another important aspect is to consider regional processes which are not considered in the global model. For example, Yin et al. (2017, HESS, Water scarcity under various socio-economic pathways and its potential effects on food production in the Yellow River Basin) applied the global models in the Yellow River basin. They used the water regulation rule, which is currently adopted by water management practices in the river basin, to redistribute surface water sources among the sub-river basins. This will facilitate the use global models for regional application.

Page 19 Ln 4 'research community to become a...science...'. Although I agree with this statement, the sentence could be rewritten.

Page 27 Ln 13, The paper has been formally published. The title of the paper has been updated as 'Intercomparison of global river discharge simulations focusing on dam operation—multiple models analysis in two case-study river basins, Missouri–Mississippi and Green–Colorado'.

Page 37 Table 1, Although DBH model has considered plant physiological processes, it is not a Dynamic Vegetation Model. DBH should be classified as 'Land Surface Model'.

Page 38 Figure 1, The impression of the Groundwater figure (left panel) is that Area In Drought (AID) to groundwater is smaller with human impacts than with natural condition in drought year 2014. We expect that groundwater drought may be smaller under natural condition than actual condition with human impacts. Perhaps here groundwater is water in shallow groundwater pool which does not include deep groundwater aquifers. It may be useful to clarify the exact meaning of the figure.

Qiuhong Tang

---

## Referee Comment (RC2) · T. Zhou (Referee) · 7 Jun 2017

The manuscript "Human-water interface in hydrological modeling: Current status and future directions" By Wada et al. reviews the current and future research directions about modeling human impacts on water resources. Generally this manuscript is well-written and includes a lot of information. I would recommend publication after considering the following minor issues.

The structure of this manuscript could be improved for a better flow. From the way it's organized, it looks like the "current status" is discussed in Section 2 and the "future directions" is in Section 3. However in fact they are mixed in these sections. In my opinion, Section 4 could be one of the subsections of Section 3 as it's also about future

directions. Reorganizing the structure or modifying the title of each section would make the paper clearer to the readers.

Title of Section 2 may change to something like "Evolution of representing human impacts in hydrological modeling" to make it more specific.

Section 3.1 is about modeling human impacts on extremes but only drought is discussed here. At least one paragraph about human activities on flood events and how does flow regulation (dams) control the flood should be included here.

Section 3.5 is about urbanization, which is fine. But a boarder review about how to represent the land use land cover change in hydrological modeling is more informative than just focusing on the urban area.

Page 13 Line 6: Using "compared" rather than "unveiled"?

At the end of Section 3.8 the authors implied that satellite-based measurements should combine with ground-based observations to reduce the uncertainties. Actually a number of works had already been exploring on this topic for over a decade which need to be mentioned here. For example, the Tropical Rainfall Measuring Mission (TRMM) Multi-satellite Precipitation Analysis (TMPA) combines products from multiple satellite and ground observations from the Global Precipitation Climatology Centre (GPCC) (Huffman et al. 2010). Some studies also evaluated the consistency between the pure satellite-based measurements (TRMM) and TMPA at regional scale (e.g. Villarini 2010) and global scales (e.g. Zhou et al. 2014).

Section 3.9 and Section 4: when discussing the socio-hydrological modeling and the interactions with policy making, it worth mentioning the Global Change Assessment Model (GCAM)(http://www.globalchange.umd.edu/gcam/), in which a long time effort has been made to incorporate different social sectors into one modeling framework, including energy, water, economy, climate, policy, and agriculture.

References:

Huffman, G. J., R. F. Adler, D. T. Bolvin, and E. J. Nelkin, 2010: The TRMM multi-satellite precipitation analysis (TMPA). Satell. Appl. Surf. Hydrol., 1–23. http://link.springer.com/10.1007/978-90-481-2915-7_1.

Villarini, G., 2010: Evaluation of the Research-Version TMPA Rainfall Estimate at Its Finest Spatial and Temporal Scales over the Rome Metropolitan Area. J. Appl. Meteorol. Climatol., 49, 2591–2602, doi:10.1175/2010JAMC2462. http://journals.ametsoc.org/doi/abs/10.1175/2010JAMC2462.

Zhou, T., B. Nijssen, G. J. Huffman, and D. P. Lettenmaier, 2014: Evaluation of Real-Time Satellite Precipitation Data for Global Drought Monitoring. J. Hydrometeorol., 15, 1651–1660, doi:10.1175/JHM-D-13-0128.1. http://journals.ametsoc.org/doi/abs/10.1175/JHM-D-13-0128.1.

---

## Referee Comment (RC3) · Pat YEH (Referee) · 7 Jun 2017

This is a very excellent review paper, written in a very comprehensive and also clear manner. No typos can be found. Also this paper has nicely included various branches of the research on the "Human-water interface in hydrological modelling", and comprehensively acknowledged almost all relevant literature published recently. The scope and width as a review paper are very good. If some potential improvements can be suggested to the authors, it would be that this paper has a slight shortcoming on its structure and hence clarity. I recommend the publication of this paper after minor re-structuring in addition to some minor editorial revisions. (See detailed review comments below) Another concern is that the impact and modeling of human-induced land use/land cover change was not mentioned in the entire paper, but that related urban-

ization was indeed included in Section 3.5. The authors may mention the rationale why this topic is not categorized as one of the "human-water interface".

Major Comments:

Regarding the structure of the main contents of this paper, Section 3, currently it has the following 9 subsections:

3.1 Modelling human impacts on extremes 3.2 Human impact modelling and indicators 3.3 Modelling human impacts on groundwater resources 3.4 Incorporating regional water management 3.5 Representing rapid urbanization 3.6 Global models for regional use 3.7 Need for model intercomparison 3.8 Observing and sharing human water management information 3.9 Modelling human activities at multiple spatial scales

Which can be categorized as two groups – Modelling purposes: 3.1, 3.2, 3.3, 3.4, 3.5 Modelling Issues: 3.5, 3.6, 3.7, 3.8, 3.9

In reviewing the relevant modelling issues, the contributions made in this paper are not as well-organized as in reviewing modelling purposes in the first 5 sub-sections. Indeed, some excellent discussions on modelling issues are given together with modelling purposes; one such example is on the bottom of page 11 (section 3.5). For those "common" modelling issues (irrespective of any particular modelling purpose), perhaps it is better to move to a new Section 4 to summarize common key modelling issues, also including the use of remote sensing data and global vs. regional modelling strategy. In this new section, some of the following significant modelling issues, which have not been systematically categorised and reviewed yet, can be summarized with more depths to add the completeness of this paper:

1. Uncertainties in data – this can be largely divided into the uncertainties in (1) climatic forcing data, and (2) calibration/validation data (and data for parameter estimation/specification)

2. Model-scale issue - Given the fact that most (if not all) LHMs and LSMs reviewed

in this paper are global models, the issue arises whether these global-scale models suitable to be applied to study regional problems? Or instead, there is urgent need to develop regional model in order to resolve the sensitivities?

3. Sub-grid variability and the scaling effect: What are the optimal grid resolution for one specific modelling purpose? The algorithms used in current LHMs and LSMs are largely based on small-scale understanding. However, If the large-scale models are commonly applied, how can these generally nonlinear small-scale effects be parametrized and scaled-up to be meaningful for large-scale model applications?

4. Understanding and evidence on the mechanisms and pathways of interactions from observations. For example, it is still not clear how the urbanization (as well as other land use / land cover changes) change has caused changes in water and energy budget partitioning. Related to studies are too scarce to gain solid understanding. Therefore, it is difficult to judge whether the parametrization and parameters used and specified in the modelling tasks are realistic, and there is a lack of data to prove the models indeed capture the impacts of human activities on the water cycle.

One more comment is on the Figure 1. The explanations in the end of Section 3.1 for this figure, and also in the figure caption, are not enough for any readers to understand the entire figure, since the authors only presented the implications of some part of this figure without explaining what and how it was calculated plotted. I suggest to provide more clear explanations on different parts of this figure.

Editorial Comments:

For the spelling check of MS Word, suggest change all "modelling" into "modelling".

P2L9: "on the hydrological cycle"

P4L7: "and runoff was routed down the simulated river systems".

P4L15-19: The differentiation between the "Conceptual" and "Physically based" models have not been mentioned, so it reads pretty vague here regarding the discussion in

this paragraph. It is nicer to explain how recently LHMs are becoming more "physically based" by giving some explanations or relevant examples.

P7L5: "important water use dominated by groundwater pumping led to a. . .."

P7L10: Both references Falkenmark (1989) and Falkenmark et al (1997) are missing in the reference section.

P8L10: "be addressed within the hydrological community"

P10L28: "Another example is the long-distance and cross-basin water diversions that provides additional water supplies. . ."

Also here the recently Chinese North-Water-South-Transfer Project starting from the end of 2014 can be another well-known important example in addition to that already mentioned Periyar Project and Kurnool Cudappah Canal in India and the Irtysh-Karaganda Canal in Central Asia.

P11L3: "but also for comprehensive data collection"

P12L30-31: These two ways read very similar to each other. Suggest to explain the difference more clearly.

P13L13: "Masaki et al. (2016) was the first to compare the simulation. . ."

P14L5: "has benefited considerably from such coordinated 5 data collection and distribution efforts in the past, but it is. . ."
* * *

---

## Author Comment (AC1) · 10 Jul 2017

Please kindly find attached our detailed responses to Referee 1, 2, and 3.

Yoshihide Wada, on behalf of my co-authors

Please also note the supplement to this comment:
https://www.hydrol-earth-syst-sci-discuss.net/hess-2017-248/hess-2017-248-AC1-supplement.pdf
* * *